# Advances in Studies on the Pharmacological Activities of Fucoxanthin

**DOI:** 10.3390/md18120634

**Published:** 2020-12-11

**Authors:** Han Xiao, Jiarui Zhao, Chang Fang, Qi Cao, Maochen Xing, Xia Li, Junfeng Hou, Aiguo Ji, Shuliang Song

**Affiliations:** 1Marine College, Shandong University, Weihai 264209, China; hanxiao201936678@mail.sdu.edu.cn (H.X.); zhaojiarui@mail.sdu.edu.cn (J.Z.); caoqi@mail.sdu.edu.cn (Q.C.); xingmaochen@mail.sdu.edu.cn (M.X.); xiali@sdu.edu.cn (X.L.); houjf@sdu.edu.cn (J.H.); 2Test Center for Agri‐Products Quality of Jinan, Jinan 250316, China; cf6061@jn.shandong.cn; 3School of Pharmaceutical Sciences, Shandong University, Jinan 250012, China

**Keywords:** fucoxanthin, stability, pharmacological activities, anticancer, anti-obesity, intestinal flora

## Abstract

Fucoxanthin is a natural carotenoid derived mostly from many species of marine brown algae. It is characterized by small molecular weight, is chemically active, can be easily oxidized, and has diverse biological activities, thus protecting cell components from ROS. Fucoxanthin inhibits the proliferation of a variety of cancer cells, promotes weight loss, acts as an antioxidant and anti-inflammatory agent, interacts with the intestinal flora to protect intestinal health, prevents organ fibrosis, and exerts a multitude of other beneficial effects. Thus, fucoxanthin has a wide range of applications and broad prospects. This review focuses primarily on the latest progress in research on its pharmacological activity and underlying mechanisms.

## 1. Introduction

In the last hundred years or so, as our understanding of the ocean continues to deepen, an increasing number of scientists have focused on the natural products in the ocean. Many types of oceanic algae have received a significant amount of attention in the food, medicine, and health products industries. Among the compounds synthesized by algae, fucoxanthin, an orange non-vitamin A carotenoid, is the most abundant of about 700 natural carotenoids, accounting for more than 10% of their aggregate mass. Fucoxanthin is mostly dark yellow to brown and is involved in the photochemical system II of photosynthesis. It can be obtained from a wide range of sources, mostly from various algae, marine phytoplankton, aquatic shells, and other animals and plants. The molecular formula of fucoxanthin is C_42_H_58_O_6_, and its chemical structure is shown in the Figure 1 [1]. Fucoxanthin contains unique olefinic bonds (conjugated structure) and certain oxygen-containing functional groups, such as epoxides and hydroxyl, carbonyl, and carboxyl groups. Its molecular weight is very small and can be easily absorbed by the organism. Fucoxanthin is not toxic to human skin cells, liver, kidney, spleen, and gonadal tissue, and, therefore, can have several applications [2,3]. However, due to the presence of a large number of unsaturated bonds, fucoxanthin is rather unstable. It is well documented that fucoxanthin exhibits numerous biological effects, such as broad-spectrum anti-tumor activity. In vitro, it shows anti-proliferation effects on a variety of cancer cell lines. In addition, fucoxanthin was shown to promote weight loss. This is a relatively innovative discovery in the previous natural medicines. Fucoxanthin has also been found to interact with the intestinal flora and is capable of alleviating oxidative stress. In addition, it has anti-inflammatory, anti-fibrotic, antibacterial, neuroprotective, anti-hyperuricemia, anti-depression, and other activities. This review is mostly focused on the sources, biological activity, and mechanism of action of fucoxanthin.

## 2. Different Sources of Fucoxanthin and Stability Improvement

### 2.1. Source of Fucoxanthin

In the past, fucoxanthin was obtained mostly from large brown algae such as kelp, wakame, and hijiki [4]. However, these macroalgae are mainly cultured in Asia, and they contain a very low amount of fucoxanthin [5]. The concentration of fucoxanthin in microalgae is markedly higher than in macroalgae [6], the tolerance for salt-alkali conditions is also higher, and the single-cell structure of microalgae results in more efficient from photosynthesis can be used in the growth and reproduction process [7], and the technology of the industrial production of microalgae is continuously developing. However, although there are many species of oxygen-enriched xanthine producing microalgae, only a few of them are used commercially to produce fucoxanthin [8,9]. The fucoxanthin content in seven microalgae isolates was determined, and its content was the highest in the biomass of *Synuroceae (Synphyceae)*. Another species of microalgae, *Tisochrysis lutea*, can also provide a higher yield of fucoxanthin, and because it has no cell wall, fucoxanthin can be easily extracted [10]. Therefore, microalgae will become a more marketable source of fucoxanthin in the future. A summary of the sources of fucoxanthin in pharmacological studies is shown in Table 1.

### 2.2. Improvement of the Stability of Fucoxanthin

Due to the unstable and easily oxidized properties of fucoxanthin, its application imposes some limitations to be resolved. Therefore, it is particularly important to improve the extraction process.

#### 2.2.1. Use of Emulsifier

The type and concentration of natural emulsifiers significantly affect the formulation, stability, lipid digestion, and bioavailability of fucoxanthin [11]. The use of an emulsion-based delivery system can effectively improve the bioavailability of fucoxanthin. Experimentally, gum arabic was used as an emulsifier, and γ-cyclodextrin was used as a stabilizer to prepare emulsified fucoxanthin [12]. In the emulsified preparation, the content of fucoxanthin remained above 90% after being exposed to 40 °C for 4 months, while the content of non-emulsified fucoxanthin fell to an undetectable level within one month. These results indicate that the emulsifier can improve the stability and bioavailability of fucoxanthin.

#### 2.2.2. Encapsulation of Fucoxanthin

In addition to the selection of emulsifiers, the type of encapsulation of fucoxanthin is also very important. For example, fucoxanthin can be encapsulated into oil/water emulsion by high-pressure homogenization (HPH) or straight-through microchannel emulsification (MCE) [13]. MCE ensures significantly better chemical stability than HPH at the same storage temperature. However, the release of free fatty acids (FFA) and the bioavailability of fucoxanthin in the emulsion generated by HPH are significantly higher than when MCE is used. Moreover, the complexes formed by employing prolactoglobulin and whey protein isolates with FX’s monomers, H-aggregates, and J-aggregates are superior to those formed by polymerization of FX in terms of encapsulation efficiency, physical stability, and biological accessibility [14]. In comparison with free fucoxanthin molecules, fucoxanthin encapsulated by whey protein isolate and acacia gum exhibits a more stable antioxidant activity [15]. Fucoxanthin can also be encapsulated with porous starch (PS) and halloysite nanotubes (HNTs) [16]. Xanthoin-loaded HNTs can be incorporated into PS (double encapsulation), forming the rock phycoxanthin that lasts longer and provides a more stable release effect than other forms of encapsulation. Fucoxanthin can also be encapsulated in casein-stabilized zein particles (zein-Cas) using a simple anti-solvent precipitation method [17]. At neutral pH, zein particles interact with fucoxanthin by hydrophobic forces, and the binding constant can reach 105 M-1, increasing the stability of fucoxanthin.

To further improve its efficacy after oral administration and improve its stability in the organism, simulated gastrointestinal experiments were performed. A gum arabic/gelatin microcapsule and alginate hydrogel beads (GA/GE/Alg-Beads) were shown to protect fucoxanthin when incubated in simulated gastric juice (SGF) [18]. In addition, the thermal stability of fucoxanthin can be significantly improved by coating with biopolymers, such as Ming maltodextrin (MD), gum arabic, and whey protein isolate (WPI), using spray drying. The degradation kinetics of biopolymer-encapsulated fucoxanthin demonstrated that gum arabic and Ming maltodextrin have protective effects on fucoxanthin. Fucoxanthin enclosed in microcapsules with palm stearin as the solid lipid core is more stable in light, high humidity, and elevated temperature than free fucoxanthin. These forms of biological encapsulation can effectively protect fucoxanthinin the acid environment of the stomach and increase the release of fucoxanthin in the intestine [19]. Encapsulation represents a novel and attractive development, establishing a satisfactory oral delivery system to increase the bioavailability of fucoxanthin, thus expanding its application and convenience of administration.

## 3. Pharmacological Activities

### 3.1. Anticancer Activity

Based on its effects on different types of tumor cells, fucoxanthin is currently being studied as a new anticancer compound [20]. One of the aspects of most conventional therapies that requires improvement is the selectivity of anti-tumor agents. Although these agents should have little or no side effects on normal cells, several conventional drugs have significant toxic effects on normal cells, and their activity against tumor cells is not high enough. Therefore, many researchers have conducted detailed experimental studies on the anticancer activity of fucoxanthin and concluded that this compound affects many types of tumor cells and can be used in combination with certain first-line drugs, providing a better therapeutic outcome. The following sections will discuss the activity of fucoxanthin in several high-incidence cancers.

#### 3.1.1. Anti-Liver Cancer Activity

The incidence of liver cancer in whole world is very high. Liver cancer is an extremely harmful malignant tumor that can be divided into two categories: primary and secondary. The etiology and exact molecular mechanism of the development of primary liver cancer involve multi-factor and multi-step complex processes that are not completely understood. The incidence of liver cancer is affected by the environment and diet. Nitrosamines are an important cause of liver cancer. Diethylnitrosamine usually increases the oxidation of lipids, and lipid peroxidation triggered by free oxygen radicals produces a variety of toxic metabolites, such as malondialdehyde and 4-hydroxynonenal. These metabolites directly attack DNA and induce cancer-promoting mutations [21,22]. Fucoxanthine has a clear therapeutic effect on diethylnitrosamine-induced liver cancer in rats, restoring normal body weight, serum albumin, antioxidant enzymes, liver enzymes, serum bilirubin, and markers of stress [23]. The rats that received only fucoxanthin did not show any changes. It has been found that fucoxanthin protects hepatocytes by upregitating antioxidant enzymes such as superoxide dismutase and catalase, and eliminating increased lipid peroxide metabolites in hepatocytes [24]. Another study documented that calcitrans extracts(CME)and their fucoxanthin-enriched fraction (FxRF) were cytotoxic to HepG2 liver cancer cells in a dose- and time-dependent manner, and FxRF (IC50: 18.89 mg/mL) was more effective than CME (IC50: 87.5 mg/mL) (*p* < 0.05). The work of Das and coworkers documented that 25 μM fucoxanthin reduced the viability of HepG2 cells by inhibiting the expression of cyclin D and inducing cell cycle arrest at the G0/G1 phase [25]. Together, these results show that fucoxanthin can act as an anti-liver cancer drug, and further studies are warranted.

#### 3.1.2. Anti-Gastric Cancer Activity

Cases of gastric cancer are frequent in the world, and its morbidity and mortality are the highest among malignant tumors. In recent years, the incidence of gastric cancer has been rising continuously. Chemotherapy for gastric cancer is poorly targeted, and causes significant damage to normal cells, necessitating the development of targeted drugs. Mcl-1, an anti-apoptotic member of the Bcl-2 family of proteins, has recently attracted widespread attention due to its potential as a drug target in cancer treatment [26]. Moreover, Mcl-1 is the target protein of the TRAIL signaling pathway, and the activation of TRAIL can downregulate the expression of Mcl-1 at the transcriptional and post-transcriptional levels, further indicating that Mcl-1 can be used as a target of cancer drugs [27]. The STAT3 protein is an important nuclear transcription factor that activates genes coding for inflammatory factors and tumor stimulating factors. The expression of these target genes can cause chronic inflammation and the development of tumors. Fucoxanthin reduces the expression of Mcl-1 and STAT3 in SGC-7901 and BGC-823 gastric cancer cells in a dose-dependent manner, causing cell cycle arrest in the S phase and apoptosis in the G2/M phase. At the same time, fucoxanthin is not toxic to normal cells, which is of great significance for potential application in gastric adenocarcinoma therapy. These findings were confirmed in MGC-803 cells, in which fucoxanthin can downregulate the expression of STAT3, cyclin B1, and survivin, induce cell cycle arrest in the G2/M phase, and trigger apoptosis [28]. A study focused on identifying the mechanism underlying the activity of fucoxanthin demonstrated that this compound inhibits the viability of SGC-7901 cells by upregulating the expression of beclin-1, LC3, and cleaved caspase-3 and downregulating the expression of Bcl-2 [29]. The ability of fucoxanthin to induce autophagy and apoptosis indicates that in-depth studies on its use for the targeted therapy of gastric cancer should be conducted.

#### 3.1.3. Anti-Leukemia Activity

Chronic myeloid leukemia (CML) is a malignant tumor formed by the clonal proliferation of bone marrow hematopoietic stem cells and is the most frequent type of chronic leukemia all over the world. Imatinib is routinely used to treat CML, but it produces many adverse reactions and cannot eradicate the disease-causing gene. Moreover, it requires life-long administration and is expensive. Therefore, the cytotoxicity of fucoxanthin alone and in combination with conventional anticancer drugs imatinib (Imat) and doxorubicin (Dox) was examined in vitro using two human leukemia cell lines, K562 and TK6 [30]. Fucoxanthin alone was cytotoxic to K562 cells and had an anti-proliferative effect on K562 and TK6 cells. In the co-incubation, the observed effect is mainly the effect of fucoxanthin alone. Although the anti-proliferative effect of imatinib and doxorubicin was not enhanced, it can be clearly observed that fucoxanthin affects K562 and TK6. However, the specific anti-proliferation mechanism of fucoxanthin cannot be fully explained by the induction of DNA damage or apoptosis, and additional investigation is necessary. It was also demonstrated that fucoxanthin can induce apoptosis of human acute promyelocytic leukemia cells [31]. This effect was related to the early loss of mitochondrial membrane potential, but not to the increased formation of reactive oxygen species. The specific mechanism may involve the induction of HL-60 cells through mitosis-mediated changes in membrane permeability and activation of caspase-3. Fucoxanthin can also affect human erythroleukemia HEL cells, reducing their viability, increasing cell apoptosis, and arresting the cells in the G0/G1 phase. Opening the mitochondrial membrane permeability transition channel reduces the mitochondrial transmembrane potential, downregulates the expression of the anti-apoptotic Bcl-2 gene, and upregulates the expression of pro-apoptotic Bax genes; Bcl-xL proteins form a pro-apoptotic complex and releases cytochrome C, leading to the enhanced expression of the pro-apoptotic caspase-3 gene and increased level of caspase-3 protein. These changes trigger apoptosis and are responsible for the significant anti-leukemic activity of fucoxanthin.

#### 3.1.4. Anti-Breast Cancer Activity

Breast cancer is a malignant tumor that occurs in the epithelial tissue of the breast and is a common tumor that threatens women’s physical and mental health. Previous studies in human umbilical vein endothelial cells (HUVECs) have shown that fucoxanthin has anti-angiogenic effects [32]. Fucoxanthin extracted from *Undaria pinnatifida* inhibits the proliferation, migration, and formation of tubular structures by Human Lymphatic Endothelial Cells (HLEC) [33]. Additionally, fucoxanthin can be used in combination with a conditioned medium culture system to inhibit the malignant phenotype of human breast cancer MDA-MB-231 cells and reduce tumor-induced lymphangiogenesis. These effects of fucoxanthin are mediated by a significant reduction in the levels of vascular endothelial growth factor (VEGF)-C, VEGF receptor-3, nuclear factor kappa B (NF-κB), phosphorylated Akt, and phosphorylated PI3K in HLEC. Fucoxanthin also reduced the microlymphatic vessel density (micro-LVD) in the MDA-MB-231 breast cancer model in nude mice. These experimental results show that fucoxanthin inhibits tumor-induced lymphangiogenesis in vitro and in vivo, highlighting its potential use as an anti-lymphangiogenic agent for the prevention of metastasis in the comprehensive treatment of breast cancer patients. Another study documented that fucoxanthin can also damage the integrity of the endoplasmic reticulum membrane by regulating the expression of Bcl-2 family proteins [34]. In MCF-7 cells, fucoxanthin can induce the translocation of calcium ions from the endoplasmic reticulum into the cytoplasm, eventually activating cell apoptosis. In addition to fucoxanthin, its metabolic derivative fucoxanthinol can also reduce the viability of MCF-7 and MDA-MB-231 cells in a dose- and time-dependent manner [35]. Moreover, the metabolic derivative is more effective than fucoxanthin, and it is also related to MDA-MB-231 cells and has an inhibitory effect on members of the NF-κB pathway, including p65, p50, RelB, and p52. Fucoxanthin also reduces the expression of transcription factor SOX9 at the transcriptional level in MDA-MB-231 cells. Fucoxanthin can be combined with adriamycin to selectively activate the oxidative stress-mediated apoptosis of breast cancer cells [36]. Fucoxanthin can inhibit the activation of p38, JNK, and p53 in cardiomyocytes, protecting them from lipid peroxidation and apoptosis, thereby alleviating doxorubicin-induced cardiotoxicity, so the combination of the two can achieve better side effects and less anti-breast [37]. The activity of fucoxanthin against breast cancer suggests novel strategies for the treatment of this type of tumor.

#### 3.1.5. Glioma

Glioma is the most common primary tumor of the central nervous system. Human glioma U87 and U251 cell lines are classified as the grade IV glioma, called glioblastoma multiforme (GBM) [38,39]. GBM is the most severe malignant astrocytoma that grows uncontrollably and invades neighboring brain structures and metastases to distant sites [40]. The prognosis of patients with GBM is poor since even the combination of surgical resection and radiotherapy or chemotherapy does not ensure a favorable outcome. Liu and coworkers demonstrated that fucoxanthin is toxic to U87 and U251 cell lines, but does not exert an inhibitory effect on normal neurons measured by the changes in the mitochondrial membrane potential and in the expression of apoptosis-related proteins [41]. As mentioned earlier, fucoxanthin can induce apoptosis of HL-60 cells, and the early loss of mitochondrial membrane potential leads to permeability changes and caspase-3 activation [31]. On the other hand, the level of apoptotic proteins—including cleaved PARP, caspase-3, and caspase-9—increased significantly, confirming once again that fucoxanthin induces mitochondria-mediated apoptosis to realize its inhibitory effect on glioblastoma cells. Additionally, fucoxanthin reduced the accumulation of phosphorylated Akt and mTOR proteins. Control experiments utilizing PI3K inhibitors confirmed that the activation of apoptosis is inhibited by PI3K/Akt/mTOR caused similar results in fucoxanthin-treated cells, that is, fucoxanthin can inhibit glioma cells by inhibiting the PI3K/Akt/mTOR pathway, realizing mTOR and PI3K dual inhibition, in principle, it not only inhibits the activity of mTOR, but also directly inhibits the activity of AKT, so it circumvents the feedback loop of S6K1-IRS1. Therefore, we know that fucoxanthin can effectively inhibit the development of glioma.

#### 3.1.6. Anti-Colon Cancer Activity

Colon cancer is a common malignant tumor of the digestive tract. It is the second most frequent gastrointestinal tumors. As early as in 2004, fucoxanthin was found to significantly reduce the proliferation of human colon cancer cell lines Caco-2, HT-29, and DLD-1 by inducing DNA fragmentation [42]. More recently, Terasaki and collaborators reported that fucoxanthin could effectively prevent colon cancer in the ethoxymethane-dextrorotatory sodium sulfate (AOM/DSS) carcinogenic mouse model. Fucoxanthin inhibited colon damage, and detection of neuron-like integrin β1low/-/- Caspase-3 high expression cell division increased, and promoting the anoikis pathway and inhibiting tumor growth, thereby generating resistance to colon cancer [43].

#### 3.1.7. Anti-Cervical Cancer Activity

Cervical cancer is currently the most frequent gynecological malignant tumor. Recent years witnessed a trend of an increasing rate of human papillomavirus infections, which decreased the age of onset of cervical cancer and increased its incidence [44]. Fucoxanthin has been shown to act synergistically with the tumor necrosis factor-related apoptosis-inducing ligand (TRAIL) to significantly increase the apoptosis of cervical cancer cell lines HeLa, SiHa, and CaSki cell lines [45]. This interaction prompted the development of novel cervical cancer drugs. TRAIL is a member of the TNF superfamily of cytokines and is mainly expressed in the immune system [45,46]. It selectively induces apoptosis of several types of tumor cells without affecting the function of normal cells. Apoptosis mediated by upstream signaling of the PI3K/Akt and NF-κB pathways is activated by TRAIL and inhibited by fucoxanthin. The inhibitors of PI3K and NF-κB, LY49002, and PDTC, respectively, suppressed the apoptosis of human cervical cancer cells induced by fucoxanthin or TRAIL. This finding indicates that fucoxanthin and TRAIL increase apoptosis of these cells by targeting the PI3K/Akt/NF-κB signaling pathway.

#### 3.1.8. Other Anticancer Activities

YunLong and collaborators documented that fucoxanthin inhibits the proliferation of nasopharyngeal carcinoma cells by inducing autophagy [47]. Mei and colleagues demonstrated that fucoxanthin prevents the development of non-small cell lung cancer at a dose that is relatively safe for normal cells [48]. It induces tumor cell apoptosis by regulating the expression of p53, p21, Fas, PUMA, Bcl-2, and caspase-3/8 and inhibiting the cell cycle. Also, a safe dose of fucoxanthin induces the nuclear translocation of p53 and stimulates its function as a transcriptional activator in cancer cells by causing the loss of mortalin-p53 interaction [49]. This effect reduces the level of landmark proteins related to cell proliferation, survival, and metastasis of cancer cells.

### 3.2. Anti-Obesity Activity and Its Mechanism

Obesity is a term that is often mentioned in daily life. Overweight and obesity are defined as abnormal or excessive accumulation of fat that can harm health, according to the WHO. It is a chronic metabolic disease caused by genetic and environmental factors, as well as numerous other reasons. According to the WHO data, obesity has become the world’s most common chronic disease, with more than 2 billion people affected by this condition. The most significant consequences of obesity are the abnormalities in human metabolism, increased burden on internal organs, and increased risk of fatty liver, hypertension, diabetes, coronary atherosclerosis, and other diseases. Therefore, obesity reduction and weight control are critical public health issues, and the development of safe anti-obesity drugs is urgently needed. Several lines of research indicate that fucoxanthin has very good potential to fight obesity. However, the mechanism of the anti-obesity activity of fucoxanthin is affected by many factors, including nutrition, hormones, and new elements of gross energy [50].

#### 3.2.1. Regulation of UCP1

Fucoxanthin can alleviate liver damage caused by excessive fat accumulation by regulating mitochondrial uncoupling protein 1 (UCP1), PPARγ, and C/EBPa. Typically, UCP1 is expressed only in brown adipose tissue (BAT) and represents the key molecule for metabolic heat generation. Increased expression of UCP1 causes energy expenditure to avoid excessive fat accumulation. PPARγ and C/EBPa promote the proliferation and differentiation of mature adipocytes and are the most important transcription factors in adipogenesis [51,52]. It has been shown that mice fed with fucoxanthin exhibited significantly upregulated synthesis of UCP1 protein and mRNA in white adipose tissue (WAT), and markedly reduce WAT mass. However, there was no difference in WAT quality and low UCP1 expression in mice fed sugar fat [53]. Supplementation of the diet with fucoxanthin for six weeks reduced body weight, organ weight, fat volume, and fat cell size in obese mice without affecting food intake. Moreover, fucoxanthin reduced lipid metabolism dysfunction and liver damage caused by a high-fat diet (HFD). It was also shown that fucoxanthin can downregulate PPARγ protein and upregulate UCP1 protein in the liver in a dose-dependent manner, but does not affect the level of C/EBPa protein. These effects may be responsible for the improvement of HFD-induced obesity by fucoxanthin in vivo [54]. An additional mechanism may involve the stimulation of the Sirt1/AMPK pathway. Fucoxanthin can regulate fatty acid synthesis by significantly increasing the phosphorylation of AMP-activated protein kinase (AMPK) and reducing the activity of acetyl-CoA carboxylase. It can also suppress the expression of transcription factors related to adipogenesis, including sterol regulatory element-binding protein 1c and PPARs, thereby reducing lipid accumulation in liver cells [55].

#### 3.2.2. Inhibition of α-amylase and α-glucosidase

Uncontrolled obesity in diabetic patients may further aggravate the diabetic condition. In a study related to this problem, fucoxanthin ameliorated diet-induced obesity and insulin resistance in mice, and allowed patients to maintain a normal diet without excessive caloric restriction [56]. α-glucosidase and α-amylase are essential factors regulating the digestion of starch and glucose absorption, making them a key target of research on the treatment of postprandial hyperglycemia [57]. Fucoxanthin can protect obese people from getting diabetes by inhibiting α-amylase and α-glucosidase and improving glucose oxidase activity, as was demonstrated in 3T3-L1 cells.

### 3.3. Effect on the Intestinal Flora

Intestinal flora—i.e., the normal microbes residing in the human intestine—affects body weight and digestion ability and protects against infections and autoimmune diseases. Changes in the composition of intestinal flora, its decreased diversity, and changes in metabolic pathways may affect the energy metabolism balance of the host, leading to the development or further worsening of obesity [58]. In the cecum and fecal samples of BALB/c mice after fucoxanthin administration, the ratio of fixed bacteria/Bacteroidetes, and the abundance of S24-7 and Ackerman have occurred significant changes [59]. Meanwhile, another set of experiments also confirmed that Fx could significantly inhibit the growth of obesity/inflammation-related spirulina and red algae [60]. Moreover, FX can also promote the growth of lactobacillus/lactococcus, *Escherichia coli* and some butylation-producing bacteria, thus reducing the imbalance of intestinal flora caused by HFD and inhibiting the development of obesity and related conditions. In addition, fucoxanthin can be generated in the intestine by the deacetylation of fucoxanthin by *Escherichia coli* and *Lactobacillus*, suggesting that fucoxanthin interacts with and affects *E. coli* and *Lactobacillus* in the gut, and may provide a new mechanism for its deacetylation [61]. Additionally, fucoxanthin can promote the growth of intestinal flora, and intestinal microorganisms promote the absorption of nutrients from the diet by producing enzymes that help digest certain indigestible compounds, such as xylan, cellulose, and resistant starch [62,63]. The interaction between fucoxanthin and intestinal flora is an activity that was recognized only recently and deserves particular attention. This property of fucoxanthin can be harnessed to improve the intestinal flora and maintain intestinal health.

### 3.4. Reduction in Oxidative Stress and Its Mechanism

The molecule of fucoxanthin is unstable due to its chemical structure. It has been found in relevant experiments that fucoxanthin has absorption of ultraviolet light (UVB 280–320 nm, UVA 320–400 nm) and visible light (VIS 400–700 nm). In particular, absorbs from 320 to 500 nm (UVA I to VIS, 448 nm max) and has absorbance of UVB radiation exposure (280–315 nm) [64]. It was further found that petrocyanin showed acceptable photodegradation after addition (27.5 J/cm2:5.8% UVB and 12.5% UVA absorbance), despite showing chemical photoinstability (dose 6 J/cm2:35% UVA and 21% VIS decreased absorbance). This also confirms that fucoxanthine is unstable and easily oxidized. Finally, fucosaflatoxin also significantly inhibited the formation of reactive oxygen species (ROS) in HaCaT cells, suggesting fucoxanthin has significant antioxidant activity.

The term ‘oxidative stress’ refers to the imbalance of oxidation/reduction processes in the organism, resulting in neutrophil inflammatory infiltration, increased protease secretion, and the production of a large number of oxidative intermediates. It is a negative effect generated by free oxygen radicals and is considered an important factor leading to aging and disease. In addition, severe oxidative stress can also lead to renal fibrosis. Experimentally, fucoxanthin was shown to reduce oxidative stress in liver cells [65]. The major mechanism of this effect is the stimulation of liver cell autophagy by inducing AMPK activation and inhibition of ROS production. Another mechanism is involved in liver damage caused by excessive consumption of alcohol, which reduces the level of the Nrf2 protein, leading to a significant downregulation of downstream proteins responsible for the antioxidative response. Fucoxanthin was demonstrated to increase the level of Nrf2 protein and its downstream targets—NQO1, HO-1, and GCLM protein—in a dose-dependent manner. These results indicate that fucoxanthin can minimize alcohol-induced liver damage in mice.

Fucoxanthin also prevents oxidative damage by increasing the production of reduced glutathione in cells. This process is mediated by enhancing the binding of the transcription factor Nrf2 to the antioxidant response element (ARE) in the promoter to activate the transcription of its target gene. When cells are attacked by ROS or electrophiles [66], Nrf2 dissociates from cytoskeleton-related proteins and enters the nucleus [67], where it initiates the expression of antioxidant enzymes, such as GCLC, GSS, NQO1, HO-1, GCLM, and others that are the downstream target genes of Nrf2 [68,69,70]. GCLC and GSS can catalyze the synthesis of GSH to protect cells against oxidative stress. In fucoxanthin treated cells, Nrf2 is enhanced in its ability to bind the ARE sequence, which leads to the enhanced transcriptional activity of Nrf2, and promotes the release of Nrf2 from KEAP-1 and its translocation to the nucleus to interact with the ARE sequence [71]. The expression of GSS and GCLC was further enhanced to induce the synthesis of GSH and enable it to play an antioxidant role in the organism.

Fucoxanthin can also effectively prevent oxidative damage in HepG2 cells induced by arachidonic acid (AA) and iron, and reduce the level of oxides and inflammation in the liver [65]. Excessive AA induces extremely high levels of cellular and mitochondrial ROS, adversely affecting promoting mitochondrial permeability transition [72]. Pretreatment with fucoxanthin blocked the ability of AA and iron to induce harmful effects in HepG2 and induced a significant increase in the phosphorylation of ULK1 and AMPK, and expression of the autophagy marker beclin-1. The addition of an inhibitor of AMPK reduced the beneficial effect of fucoxanthin on mitochondria and decreased the level of beclin-1 protein. These results demonstrated that fucoxanthin induced autophagy in hepatocytes by activating the AMPK-Ulk1-mTORC1 axis, and upregulated the phosphorylation of AMPK, its downstream target ACC, and essential upstream kinase LKB1. These effects inhibited the production of ROS and mitochondrial dysfunction, alleviating oxidative stress, and protecting hepatocytes. In addition to the above-indicated mechanism, fucoxanthin can eliminate lipid peroxidation metabolites from liver cells by activating SOD, CAT, and other antioxidant enzymes, providing an anti-liver cancer effect [23].

### 3.5. Anti-Fibrotic Activity of Fucoxanthin

Fibrosis is a consequence of the necrosis of parenchymal cells in an organ and an accumulation of connective tissue, leading to structural damage and functional failure of the organ. Liver fibrosis may develop into cirrhosis, severely impairing normal liver function [73,74,75]. The activation of hepatic stellate cells (HSCs) is an essential process in the development of liver fibrosis [74]. After a liver injury, HSCs transdifferentiate into myofibroblast-like (MFB) cell types [76,77]. Fucoxanthin can prevent the expression of pro-fibrosis genes induced by TGF-β1 by inhibiting the activation of SMAD3 and quiescent HSCs, thus providing an anti-fibrotic effect. The critical event in renal fibrosis is the overexpression and deposition of the extracellular matrix (ECM). The activation of cells that produce ECM, including its essential component, fibronectin, is the key to the development of renal fibrosis [78]. The transcription factor FoxO regulates the expression of various genes and controls several cell functions [79,80], contributing to the maintenance of a low level of ROS. Sirt1 controls FoxO3α activity in response to oxidative stress by regulating its deacetylation [81,82]. Excessive ROS production promotes the nuclear translocation of FoxO and its transcriptional activity [83]. Oxidative stress enhances the activity of Akt, thereby inactivating FoxO3α by phosphorylation at serine 253 [84]. Guanyu and coworkers demonstrated that fucoxanthin attenuated the high glucose (HG)-induced ROS production and ECM secretion [79]. Also, HG inhibits Sirt1 and the activation of Akt, which not only enhances the nuclear export of FoxO3α but also reduces its transcriptional activity. Fucoxanthin strongly promotes the nuclear transport and transcriptional activity of FoxO3α and upregulates the expression of MnSOD, diminishing the level of ROS and inhibiting the expression of fibronectin and collagen induced by HG. This mechanism enables fucoxanthin to significantly suppress the expression of fibronectin and collagen IV and the production of ROS, resulting in the reduction of fibrosis in diabetic nephropathy.

### 3.6. Fucoxanthin Anti-Inflammatory Activity

Inflammation is a defense response of the body to stimuli, with redness, swelling, pain, and functional disorders as the main symptoms. Inflammation can be triggered by multiple mechanisms, and any stimulus that causes tissue damage can be an inflammatory factor.

#### 3.6.1. Inflammation Caused by Lipopolysaccharide

Lipopolysaccharide (LPS) binds its receptor, CD14, forming immune complexes, which then interact with Toll-like receptors (TLR) to promote intracellular signal transduction and release various inflammatory mediators [85]. LPS is also an endogenous mediator that induces fever [86]. Fucoxanthin has an antioxidative and anti-inflammatory effect on LPS-induced RAW 264.7 cells. In animal studies, fucoxanthin has been shown to reduce symptoms of diabetes and improve sperm production and male reproduction. The main mechanism is that fucoxanthin inhibits NO and PGE2 by inhibiting the expression of iNOS protein and the transcription of COX-2 mRNA [75]. Additionally, fucoxanthin suppresses in a dose-dependent manner the stimulation of TNF-α and IL-6 by LPS. The production and mRNA expression, fucoxanthin can also markedly reduce the loss of cell viability and mitochondrial membrane potential induced by LPS [87]. In another experimental study, fucoxanthin significantly inhibited the upregulation of IL-1β, TNF-α, iNOS, and COX-2, thereby slowing down the inflammation mediated by LPS-stimulated macrophages. Another mechanism of fucoxanthin activity relies on suppressing the phosphorylation of IκB-α, MAPKs, and Akt, which inactivates NF-κB and inhibits the LPS-induced secretion of NO and PGE2 by macrophages [88]. MAPKs participate in the pro-inflammatory signal transduction cascade and the regulation of iNOS and COX-2 by activating NF-κB in immune cells stimulated by LPS. Akt acts upstream of IKK/NF-jB activation [89]. In vivo studies utilizing the mouse model of LPS-induced sepsis demonstrated that fucoxanthin reduces inflammatory cytokines, including IL-6 and IL-1β, by regulating the NF-κB signaling, thus reducing inflammation [85]. After the ingestion of a large amount of alcohol, LPS can stimulate TLR4, generating downstream signals that activate transcription factors through intracellular signaling pathways and promote inflammation, leading to alcoholic liver damage [90,91,92,93]. The administration of a combination of fuciformin and fuciformin polysaccharide also significantly inhibited the TLR4 signaling pathway, resulting in the reduction in serum ALT in serum and alleviation of alcohol-induced liver injury [92,93].

#### 3.6.2. Anti-Dermatitis Activity

The combination of fucoxanthin and rosmarinic acid (RA) can reduce UVB-induced apoptosis of HaCaT keratinocytes by downregulating inflammasome components (such as NLRP3, ASC, caspase-1, IL-1β), effectively exerting antioxidant and anti-inflammatory effects, and preventing ROS formation and cell death [94]. The expression of antioxidant genes such as Nrf2 and HO-1 was also increased by fucoxanthin. Fucoxanthin-containing cream can effectively improve tissue plasminogen activator (TPA)-induced hyperplasia and UVB-induced acute erythema in hairless mice, resulting in reduced skin edema, epidermal thickness, MPO activity, and COX-2 expression, thus providing a satisfactory therapeutic effect [95]. Chen and collaborators found that fucoxanthin pretreatment can alleviate the symptoms of UVB-induced corneal denervation and trigeminal neuralgia by promoting the expression of Nrf2 [96]. In addition to skin inflammation caused by UV radiation, atopic dermatitis (AD) also has a high incidence worldwide. The effect of fucoxanthin on NC/Nga mice (an animal model of AD) was compared with tacrolimus (TAC) [97]. When the treatment with TAC is not adequate to improve some symptoms of AD, topical treatment with fucoxanthin can normalize the immune response by affecting regulatory innate lymphoid cells, thereby exerting an anti-inflammatory effect on keratinocytes and alleviating the symptoms of AD.

### 3.7. Other Activities

In addition to anticancer, anti-obesity, regulation of intestinal flora, antioxidative, anti-inflammatory, and other main effects, fucoxanthin possesses anti-fibrotic, antibacterial, neuroprotective, antihyperuricemic, antidepressive, and other activities.

As discussed earlier, fucoxanthin can interact with the intestinal flora. Additional studies documented the effect of fucoxanthin on 13 species of bacteria growing under aerobic conditions, and the impact was much higher on Gram-positive than Gram-negative bacteria [61,98]. Fucoxanthin can also inactivate enzymes regulating *Mycobacterium tuberculosis* cell wall, UDP-galactopyranose mutase (UGM) and arylamine-N-acetyltransferase (TBNAT), inhibiting the growth of these bacteria. Fucoxanthin has a therapeutic effect on Parkinson’s disease by binding to and antagonizing dopamine D3 and D4 receptors. Additionally, fucoxanthin exhibits multiple neuroprotective effects [99,100].

The combination of fucoidan (Fc) and fucoxanthin (Fx) inhibits the activity of xanthine oxidase (XO) in the liver and regulates the expression of proteins (GLUT9 and URAT1) related to the uric acid (UA) transporter in the kidney, reducing the serum level of UA and preventing hyperuricemia [101]. Fucoxanthin can significantly reverse the depression-like behavior of mice induced by lipopolysaccharide (LPS) [102]. This effect is mediated by the regulation of the AMPK-NF-κB signaling pathway. Fucoxanthin inhibits LPS-induced pro-inflammatory cytokines IL-1β, IL-6, and TNF-α and suppresses the overexpression of iNOS and COX-2 in the hippocampus, frontal cortex, and hypothalamus.

### 3.8. Main Metabolites and Their Activities of Fucoxanthin

In addition to the obvious anti-cancer effect of fucoxanthin, its derivatives after metabolism in the body have also become a new research focus in recent years. In recent years, studies have shown that it can be esterified or hydrolyzed to fucoxanthinol in the gastrointestinal tract and further converted into amarouciaxanthin A in the liver [103]. Specific experiments have confirmed that fucoxanthin and its derivatives fucoxanthinol and amarouciaxanthin A can be measured in the plasma, red blood cells, liver, lung, kidney, heart, spleen, and adipose tissue of mice after oral administration of fucoxanthin. Through experiments, it is clear that fucoxanthin accumulates in the heart and liver in the form of fucoxanthinol, and accumulates in fat in the form of aurixanthine A in adipose tissue [104]. This result was also verified in another experiment, fucoxanthin accumulates in the form of p-methoquinone in the mice, and fucoxanthinol and amarouciaxanthin A are also detected [105]. Meanwhile, in the follow-up experiments of Hashimoto Takash, they found that fucoxanthinol was detected in human plasma, but amarouciaxanthin A, a liver metabolite of fucoxanthin could not be detected. This also indicates that the bioavailability and metabolism of fucoxanthin are different between human subjects and mice [106]. All these indicate that the biological activity of fucoxanthin can be metabolized into fucoxanthinol and amarouciaxanthin A to exert its effects.

In terms of the specific structures of these two metabolites, the structure of fucoxanthinol is similar to that of fucoxanthin. There is a chemically active 5,6-epoxy and non-reactive 5,6-epoxy at both ends of its rigid all-trans long chain. Due to the special epoxy and propylene bond structure, fucoxanthinol have a variety of biological activities, especially in anti-cancer activity, even stronger than fucoxanthin [107]. This activity has also been confirmed by related literature, fucoxanthinol also has significant therapeutic effects on colon cancer cell lines such as Caco-2, WiDr, SW620, HCT116, and DLD-1 [108]. At the same time, amarouciaxanthin A has also been confirmed to be fully synthesized through experiments, through the stereoselective Wittig reaction of C15-allenic and C15-acetylenic tri-n- butylphosphonium salts with the unprecedented C25-3,8-dihydroxy-5,6-epoxyapocarotenal have been completed [109], the specific molecular structure of the fucoxanthinol and amarouciaxanthin A is shown in Figure 2. In addition, the two metabolic derivatives of fucoxanthin also have a variety of activities. According to literature reports, fucoxanthinol and Amarouciaxanthin A in 3T3-L1 adipocytes could show good anti-inflammatory effects by downregulating the mRNA expression of pro-inflammatory mediators and chemokines induced by RAW264.7 cells [105]. Moreover, it was found in previous reports that fucoxanthin has the effect of anti-obesity and inhibiting the differentiation of 3T3-L1 adipocytes. In the specific study of Mi-Jin Yim et al., it was found that this is mainly due to the fucoxanthin metabolism to fucoxanthinol and Amarouciaxanthin A, thus playing a further role. Fucoxanthinol can downregulate PPAR (1) mRNA expression, while amarouciaxanthin A can downregulate PPAR (1) and C/EBPr mRNA expression to inhibit the differentiation of 3T3-L1 adipocytes. In addition, they investigated the regulatory effects of amarouciaxanthin A on the mRNA and protein expression of transcriptional factors, PPARγ and C/EBPR, and on mRNA expression of markers for adipocyte differentiation, it is found that the inhibitory effect of the amarouciaxanthin A is more dominant [110]. In summary, we can understand that not only fucoxanthin, but also its metabolites have various activities that are worthy of our in-depth study and exploration.

**Table 1 marinedrugs-18-00634-t001:** Summary of the sources of fucoxanthin in pharmacological studies.

Mechanism	Source	Experimental Application	Reference
	Purchased from BGG Japan 11 Nichiyu 10 Mitsubishi Chemical Food	Emulsified powder containing 1.1% (W/W) fucoxanthin	[12]
	Purchased from BGG-Japan Co., Ltd. (Tokyo, Japan)	Investigated the formulation and stability characteristics of monodisperse oil-in-water (O/W) emulsions encapsulating fucoxanthin	[13]
	Purchased from Beijing Solarbio Science & Technology Co., Ltd. (Beijing, China)	To prepare the complex of whey protein and monomer fucoxanthin	[14]
	Extract from dry wakame from Shandong Jiayi Aquatic Food Co. LTD	Preparation of fucoxanthin nanocomposite	[15]
Enhanced fucoxanthin stability	Extract from *Sphagnum angustifolium*, purchased from Algae Resource Development Technology Company (Shiraz, Iran)	Encapsulated fucoxanthin with PS, HNT	[16]
	Purchased from Sigma-Aldrich Corp. (St. Louis, MO, USA)	Encapsulate fucoxanthin (FX)	[17]
	Purchased from YIGEDA Bio-Tech Co., Ltd. (Beijing, China)	To determine the activity of fucoxanthin in gastric acid	[18]
	Extracted from dried *Undaria pinnati**f**ida* was supplied by Jiayi Aquatic Products Co. Ltd. (Shandong, China)	In order to spray-dry the fucoxanthin coating	[19]
	Purchased from Wako Pure Chemical Co., Ltd. (Japan)	It was given intraperitoneally to the rats	[23]
	Extracted from the *Chaetoceros calcitrans*	Deal with HepG2 cancer cell	[24]
	Extracted from the brown sea algae *Lonicera aponica* (kombu)	Deal with HepG2 cancer cell	[25]
	Extracted from the brown sea algae *Undaria pinnatifida* (Wakame), provided by the company of Wuhan Heli	Deal with human gastric adenocarcinoma MGC-803 cells	[28]
	Extracted from the Japanese brown algae *Undaria pinnatifida*, supplied by Nippon Seisakusho Co. Ltd. (Beijing, China)	Deal with human gastric cancer SGC-7901cells	[29]
	Purchased from Sigma-Aldrich (St. Louis, MO, USA)	Deal with human leukemia cell lines, K562 and TK6	[30]
	It did not publicly identify the source	Deal with human promyelocytic leukemia cells	[31]
Antitumor	Extracted from the freeze-dried powder of marine alga, *Undaria pinnatifida* and *Codium fragile*, respectively	Deal with human umbilical vein endothelial cells (HUVECs)	[32]
	Extracted from *Undaria. pinnatifida*	Deal with lymphatic endothelial cells (LEC)	[33]
	Purchased from Sigma-Aldrich (St. Louis, MO, USA)	Deal with human MCF-7 breast cancer cells	[34]
	Purchased from Wako Chemicals (Richmond, VA, USA)	Deal with human breast cancer line MCF-7 and MDA-MB-231	[35]
	Purchased from Sigma-Aldrich (St. Louis, MO, USA)	Deal with human leukemia cancer cell line U937 and K562	[37]
	Purchased from Sigma Chemical Company (St. Louis, MO)	Deal with human glioma cancer cell line U87 and U251	[38]
	Extract from the edible seaweed *Undaria pinnatifida*	Deal with human colon cancer cell lines, Caco-2, HT-29, and DLD-1	[42]
	Purchased from the Oryza Oil & Fat Chemical Co. Ltd. (Aichi, Japan) and Dr. Hayato Maeda (Hirosaki University, Japan)	Deal with azoxymethane-dextrane sodium sulfate (AOM/DSS) carcinogenic model mice	[43]
	Extract from the chloroplasts of brown sea-weeds	Deal with human cervical cancer cell lines HeLa, SiHa, and CaSki	[45]
	Purchased from Sigma- Aldrich (St. Louis, MO)	Deal with Human NPC cell line C666-1	[47]
	Extract from *Laminaria japonica*	Deal with human non-small lung cancer cell	[48]
	Extract from *Undaria pinnatifida*	Deal with A549, DLD-1, H1299, MCF7, MDA-MB-231, MRC5, SKOV3, TIG-3, and U2OS cell lines	[49]
	Extract from dried powder of seaweed (*Undaria pinnatifida*) after removing carbohydrate and protein was obtained from Riken Vitamin (Tokyo, Japan)	Fed to Male Wistar rats and female KK-Ay mice	[53]
	Extract from the *Phaeodactylum tricornutum* strain UTEX 640 (SAG 1090-1b),which was obtained from the culture collection of Algae (SAG) from the University of Goettingen (Germany)	Fed to C57BL/6J mice	[54]
Anti-Obesity Effect	Purchased from Sigma-Aldrich (St. Louis, MO, USA)	Deal with FL83B cell line	[55]
	Purchased from Sigma–Aldrich (St. Louis, MO, USA)	Deal with 3T3-L1 preadipocyte cell	[56]
	Purchased from Sigma-Aldrich (St. Louis, MO, USA)	Deal with 3T3-L1 preadipocyte cell	[57]
	Purchased from Beijing HFK Bioscience Co., 90 Ltd.	Fed to BALB/c mice	[59]
	Dried *Undaria pinnati**fida* was supplied by Jiayi Aquatic Products Co. Ltd. (Shandong, China)	Fed to C57BL/6J mice	[60]
Effect on the intestinal flora	Extract from *Undaria pinnatifida*	Fed to *Escherichia coli* and *lactobacilli*	[61]
	Extract from *D**iatoma* *anceps* (wet) at the Punta Plaza location—Antarctic Continent	Deal with 3T3 mouse fibroblast	[64]
Reduction in oxidative stress	Purchased from Sigma-Aldrich (St. Louis, MO, USA)	Deal with raw 264.7 cell	[65]
	Purchased from Shandong Jiejing Group Corporation (Rizhao, Shandong, China)	Fed to Male ICR mice	[67]
	Purchased from Santa Cruz Biotechnology (Santa Cruz, CA, USA)	Deal with the human keratinocyte cell line HaCaT	[71]
	Purchased from Sigma (St. Louis, MO)	Deal with LX-2 cells	[73]
Anti-fibroti	Purchased from Biopurify (Chengdu, China)	Deal with Glomerular mesangial cells (GMCs)	[79]
	Extract from dried *Chaetoceros calcitrans*	Deal with the human liver cancer cells (HepG2)	[85]
	Extract from *Conticribra weissflogii* ND-8	Deal with C57BL/6 mice and RAW 264.7 cells	[86]
Anti-inflammatory	Purchased from Sigma-Aldrich (St. Louis, MO, USA)	Deal with Raw 264.7 cell	[88]
	Purchase from HiQ Marine Bio- tech Company in Taiwan	Deal with 70 patients who visited the outpatient department of Gastroenterology and Hepatology, with ages ranging from 20 to 75 years old	[94]
	Purchased from Sigma-Aldrich (St. Louis, MO, USA)	Deal with human immortalized keratinocytes HaCaT	[95]
	Purchased from Sigma-Aldrich (St. Louis, MO, USA)	Deal with the THP-1 human monocytic leukemia cell line and HaCaT human keratinocytes, and fed with female Swiss CD-1 mice	[96]
Anti-dermatitis	Purchased from Sigma-Aldrich (St. Louis, MO, USA)	Fed to male SD rats	[97]
	Purchased from Phytolox (Okinawa, Japan)	Deal with WEHI-3 cells and fed to male Nc/Nga mice	[98]
	Extract from *Undaria pinnatifida*	Deal with Transfected Chinese hamster ovary (CHO) cells, rat basophil leukemia (RBL) cells, U373 cells, and BA/F3 cells	[100]
	Purchased from Sigma-Aldrich (St. Louis, MO, USA)	Deal with Rat pheochromocytoma cells PC-12	[101]
Other activities	Extract from *Laminaria japonica* and purchase from Hi-Q Marine Biotech International Ltd. (New Taipei City, Taiwan)	Fed to male Sprague Dawley (SD) rats	[102]
	Purchased from Sigma-Aldrich (St. Louis, MO, USA)	Fed to Male ICR mice	[103]

## 4. Summary and Outlook

This article introduces the source, activity, and mechanism of function of fucoxanthin. Fucoxanthin is a small molecule substance derived from marine algae. It has a wide range of activities against human diseases, such as cancer, obesity, dysregulation of intestinal flora, oxidative stress, and inflammation. Importantly, fucoxanthin is not cytotoxic. However, its chemical properties and ability to be easily oxidized should be considered in its therapeutic applications. Weight loss effects of natural non-toxic products such as fucoxanthin represent a promising research direction. If the mechanism fucoxanthin-mediated weight loss can be comprehensively explained and lead to commercial production, it may become a novel, safe, and effective weight loss product. This review also summarizes the information on the mutual regulation of fucoxanthin and intestinal flora. New ideas were proposed for regulating intestinal flora and maintaining intestinal health, providing inspiration for the industrial production of fucoxanthin derivatives. It is hoped that in the future, the applications of fucoxanthin will be further developed, its possible other activities will be verified experimentally, the mechanisms that are not yet clear will be elaborated in more detail, and breakthroughs will be made to overcome its instability and low oral absorption rate. Applications of fucoxanthin will be developed more extensively and in-depth to make better use of its various biological activities in their lives, allowing it to play a greater role in the healthcare and pharmaceutical markets.

## Figures and Tables

**Figure 1 marinedrugs-18-00634-f001:**
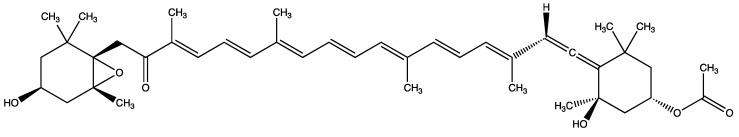
Molecular formula of fucoxanthin.

**Figure 2 marinedrugs-18-00634-f002:**
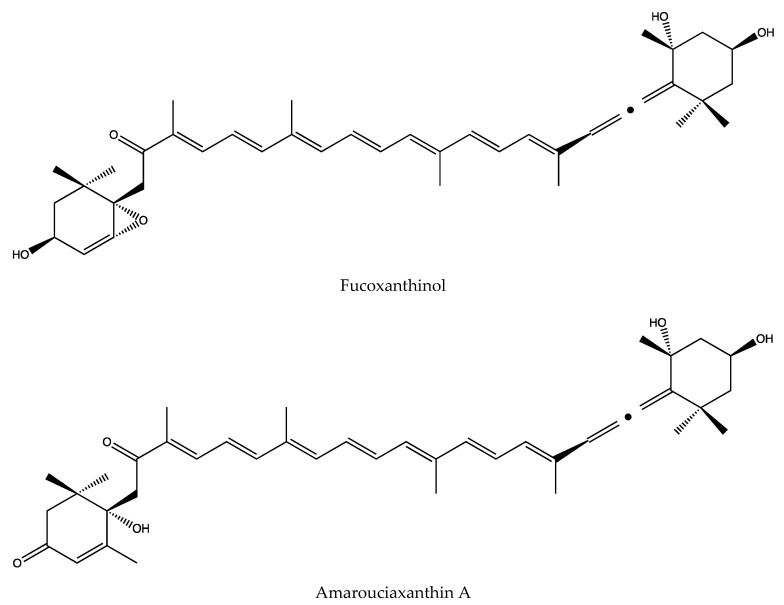
Molecular formula of fucoxanthinol and amarouciaxanthin A.

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
