# Peer review of "Advances in Studies on the Pharmacological Activities of Fucoxanthin"

_marinedrugs, 2020, doi:10.3390/md18120634_

Round 1
Reviewer 1 Report
General comments
The manuscript entitled “Advances in studies on pharmacological activities and mechanisms of fucoxanthin” presents a review on the latest knowledge on the application of fucoxanthin for pharmaceutical purposes. The title mentions “mechanisms” but doesn’t say of what. Please revise the title.
Nutraceuticals are receiving increasing attention from researchers the literature reflects that attention. This revision has scientific interest in that it offers a summary on the present knowledge of fucoxanthin and its possible applications to improve human health. However, I recommend a revision of the text so that it conveys the general message in a more careful way (the authors are sometimes careless in their sentences). There are many instances of scientific inaccuracies that should be corrected. It is incorrect to say e.g. “fucoxanthin …. is involved in the photochemical system II of photosynthesis”, when in fact, fucoxanthin is associated with PS II and participates in photosynthesis through its ability to harvest light.
The authors should always reference their information since this is a revision and the authors did not generate the information themselves. There are numerous sentences not attributed to any reference, and that must be corrected. There should be a space between the last word and the first parenthesis of the references. There are many small typing errors throughout the text, which should be thoroughly revised.
Fucoxanthin is abbreviated as Fx and FX, please use the same abbreviation. In carotenoid studies fucoxanthin is often abbreviated as “Fuco” or not at all abbreviated. I don’t see the advantage of using abbreviation here.
Detailed comments
I have made some changes to the abstract as an example of the necessary revision of text.
Abstract: Fucoxanthin is carotenoid present in many species of marine brown macro-algae and both freshwater and marine micro-algae. It is characterized by a small molecular weight, has diverse biological activities and can be easily oxidized, thus protecting cell components from ROS. Fucoxanthin inhibits the proliferation of a variety of cancer cells, promotes weight loss, acts as an antioxidant and anti-inflammatory agent, interacts with the intestinal flora to protect intestinal health, prevents organ fibrosis, and exerts a multitude of other beneficial effects. Thus, fucoxanthin has a wide range of applications and broad prospects. This review focuses primarily on the latest progress in research on its pharmacological activity and underlying mechanisms.
In section 2.2. the information on fucoxanthin the “epoxide nature of fucoxanthin” is a repetition from the introduction.
The first sentence in the introduction should be changed to convey the interest in the application of products from marine organisms, since scientists have been devoting their attention to the natural products of the oceans for centuries. As an additional example, in sentence 26, although I understand what the authors mean by “fucoxanthin, an orange non-vitamin A carotenoid” it is scientifically incorrect. In fact, fucoxanthin is a carotenoid, which is not a precursor of vitamin A, unlike e.g. b-carotene. That should be corrected.
I understand the purpose of mentioning the seriousness of the various types of cancer in China, yet is it really necessary to restrict their importance to China alone?
Line 187. The name of the species Undaria pinnatifida should be in italic and the authors should say what type of alga it is. Please use italic also for Escherichia coli and Lactobacillus, later in the text.
Line 201. What is the “metabolic derivative” of fucoxanthin?
Line 221. This sentence makes little sense, please revise “Enhanced mitochondrial membrane permeability results in the release of cytochrome c the cytoplasm, increase in the Bax/Bcl-2 ratio, and induction and execution of caspase cascade.”
Line 263. What is a “safe dose” of fucoxanthin? Why is it important, any side effects?
Line 268. Please use the official definition of obesity. WHO - Overweight and obesity are defined as abnormal or excessive fat accumulation that may impair health. See https://www.who.int/news-room/fact-sheets/detail/obesity-and-overweight.
Line 302. Why is physalxanthin relevant for the present review and, if relevant, what is it?
Line 330. Since you mention the absorption spectra of fucoxanthin, please state the absorbed wavelengths and not the wavelengths of UV and visible light (that is irrelevant to know which lambda is absorbed by fucoxanthin). And why call it “fuco yellow”? See https://www.int-res.com/articles/meps/38/m038p259.pdf
Line 333. What is fucosaflatoxin and why do you mention it?
Line 346. What is “Fucoxanthin Fx”?
Line 364. What “mitochondrial respiration”?
Line 464. The authors claim that “This article introduces the source, activity, and mechanism of function of fucoxanthin. Fucoxanthin is a small molecule substance derived from marine algae.” I believe this article revises the available information on the sources (although little is said about the possible sources) and pharmacological activities of fucoxanthin. Additionally, fucoxanthin is not “derived”, it is obtained, mostly through extraction, from macro- and micro-algae. The authors should be carefull in their summary and should revise it.
I've added further comments directly on the pdf file.

Author Response
Dear reviewer,
At first, thank you very much for your advice on this article. Secondly, I would like to express my apologies for the imperfect writing of this article, which has brought you a bad experience in reading. The following is my reply to your suggestions, if there are any incorrect points, I hope you can point them out and I will correct them in time.
Kind regards,
Han Xiao Shan Dong University.
Point 1: The manuscript entitled “Advances in studies on pharmacological activities and mechanisms of fucoxanthin” presents a review on the latest knowledge on the application of fucoxanthin for pharmaceutical purposes. The title mentions “mechanisms” but doesn’t say of what. Please revise the title.
Response 1: The title has been modified
Point 2: Nutraceuticals are receiving increasing attention from researchers the literature reflects that attention. This revision has scientific interest in that it offers a summary on the present knowledge of fucoxanthin and its possible applications to improve human health. However, I recommend a revision of the text so that it conveys the general message in a more careful way (the authors are sometimes careless in their sentences). There are many instances of scientific inaccuracies that should be corrected. It is incorrect to say e.g. “fucoxanthin …. is involved in the photochemical system II of photosynthesis”, when in fact, fucoxanthin is associated with PS II and participates in photosynthesis through its ability to harvest light.The authors should always reference their information since this is a revision and the authors did not generate the information themselves. There are numerous sentences not attributed to any reference, and that must be corrected. There should be a space between the last word and the first parenthesis of the references. There are many small typing errors throughout the text, which should be thoroughly revised.
Response 2: The formatting issues in the article have been modified
Point 3: Fucoxanthin is abbreviated as Fx and FX, please use the same abbreviation. In carotenoid studies fucoxanthin is often abbreviated as “Fuco” or not at all abbreviated. I don’t see the advantage of using abbreviation here.
Response 3: The abbreviation for fucoxanthin has been corrected
Point 4: I have made some changes to the abstract as an example of the necessary revision of text.Abstract: Fucoxanthin is carotenoid present in many species of marine brown macro-algae and both freshwater and marine micro-algae. It is characterized by a small molecular weight, has diverse biological activities and can be easily oxidized, thus protecting cell components from ROS. Fucoxanthin inhibits the proliferation of a variety of cancer cells, promotes weight loss, acts as an antioxidant and anti-inflammatory agent, interacts with the intestinal flora to protect intestinal health, prevents organ fibrosis, and exerts a multitude of other beneficial effects. Thus, fucoxanthin has a wide range of applications and broad prospects. This review focuses primarily on the latest progress in research on its pharmacological activity and underlying mechanisms.
Response 4: The abstract has been modified
Point 5: In section 2.2. the information on fucoxanthin the “epoxide nature of fucoxanthin” is a repetition from the introduction.
Response 5: In section 2.2, the redundant content has been deleted
Point 6: The first sentence in the introduction should be changed to convey the interest in the application of products from marine organisms, since scientists have been devoting their attention to the natural products of the oceans for centuries. As an additional example, in sentence 26, although I understand what the authors mean by “fucoxanthin, an orange non-vitamin A carotenoid” it is scientifically incorrect. In fact, fucoxanthin is a carotenoid, which is not a precursor of vitamin A, unlike e.g. b-carotene. That should be corrected.
Response 6: The error in the introduction has been corrected, and the incorrect expression in sentence 26 has also been corrected.
Point 7: I understand the purpose of mentioning the seriousness of the various types of cancer in China, yet is it really necessary to restrict their importance to China alone?
Response 7: When I mention China here, I just want to emphasize the importance of cancer treatment in China. I realize that there are limitations in my expression, and I have deleted this limited view.
Point 8: Line 187. The name of the species Undaria pinnatifida should be in italic and the authors should say what type of alga it is. Please use italic also for Escherichia coli and Lactobacillus, later in the text.
Response 8: Line 187, a typo in the name of the strain has been corrected.
Point 9: Line 201. What is the “metabolic derivative” of fucoxanthin?
Response 9: The fucoxanthin derivative i want to write is fucoxanthinol.
Point 10: Line 221. This sentence makes little sense, please revise “Enhanced mitochondrial membrane permeability results in the release of cytochrome c the cytoplasm, increase in the Bax/Bcl-2 ratio, and induction and execution of caspase cascade.”
Response 10: Line 221, this sentence has been modified.
Point 11: Line 263. What is a “safe dose” of fucoxanthin? Why is it important, any side effects?
Response 11: Line 263, the safe dose refers to the dose that is relatively safe to use compared with normal cells, which is described in the cited literature without specific explanation. However, according to studies reported in other literatures, fucoxanthine has no toxicity to normal cells.
Point 12: Line 268. Please use the official definition of obesity. WHO - Overweight and obesity are defined as abnormal or excessive fat accumulation that may impair health. See https://www.who.int/news-room/fact-sheets/detail/obesity-and-overweight.
Response 12: Line 268, I have revised it to the authoritative interpretation of WHO.
Point 13: Line 302. Why is physalxanthin relevant for the present review and, if relevant, what is it?
Response 13: Line 302, phosphoflavin has nothing to do with this article. Sorry for my miswriting here.
Point 14: Line 330. Since you mention the absorption spectra of fucoxanthin, please state the absorbed wavelengths and not the wavelengths of UV and visible light (that is irrelevant to know which lambda is absorbed by fucoxanthin). And why call it “fuco yellow”? See https://www.int-res.com/articles/meps/38/m038p259.pdf
Response 14: Line 330, according to the article I read, Fucoxanthin absorbs from 320 to 500 nm (UVA I to VIS, 448 nm max). It has absorbance of UVB radiation exposure (280–315 nm). I am very sorry that according to the literature I read, the author only confirmed that fucoxanthin can improve the skin damage caused by UVB radiation, and did not mention the specific maximum uv absorption wavelength.
Point 15: Line 333. What is fucosaflatoxin and why do you mention it?
Response 15: Line 333, fucaflatoxin is a handwriting error and has been corrected
Point 16: Line 346. What is “Fucoxanthin Fx”?
Response 16: Line 346, fucoxanthin Fx is a handweiting error and has been corrected
Point 17: Line 364. What “mitochondrial respiration”?
Response 17: Line 364, mitochondrial respiration was a handwriting error, which should have been mitochondrial high-permeability transition
Point 18: Line 464. The authors claim that “This article introduces the source, activity, and mechanism of function of fucoxanthin. Fucoxanthin is a small molecule substance derived from marine algae.” I believe this article revises the available information on the sources (although little is said about the possible sources) and pharmacological activities of fucoxanthin. Additionally, fucoxanthin is not “derived”, it is obtained, mostly through extraction, from macro- and micro-algae. The authors should be carefull in their summary and should revise it.
Response 18: Line 464, there is the supplement: Sources of fucosanthin in different experiments. ( The form cannot be pasted here. I have attached the specific form to the revised article and uploaded it.)
|
Mechanism |
source |
experimental application |
Reference |
|
|
Purchased from BGG Japan 11 Nichiyu 10 Mitsubishi Chemical Food |
Emulsified powder containing 1.1% (W/W) fucxanthin |
12 |
|
|
Purchased from BGG-Japan Co., Ltd. (Tokyo, Japan) |
investigated the formulation and stability characteristics of monodisperse oil-in-water (O/W) emulsions encapsulating fucoxanthin |
13 |
|
|
Purchased from Beijing Solarbio Science & Technology Co., Ltd. (Beijing, China) |
To prepare the complex of whey protein and monomer fucxanthine |
14 |
|
|
Extract from dry wakami from Shandong Jiayi Aquatic Food Co. LTD |
Preparation of fucoxanthin nanocomposite |
15 |
|
Enhanced fucxanthin stability |
Extract from S. angustifolium ,purchased from Algae Resource Development Technology Company (Shiraz, Iran) |
Encapsulated Fucoxanthin with PS, HNT |
16 |
|
|
Purchased from Sigma-Aldrich Corp. (St. Louis, MO, USA) |
encapsulate fucox- anthin (FX) |
17 |
|
|
Purchased from YIGEDA Bio-Tech Co., Ltd (Beijing, China) |
To determine the activity of fucoxanthine in gastric acid |
18 |
|
|
Extracted from dried Undaria pinnatida was supplied by Jiayi Aquatic Products Co. Ltd. (Shandong, China) |
In order to spray - dry the fucoxanthin coating |
19 |
|
|
Purchased from Wako Pure Chemical Co., Ltd. (Japan) |
It was given intraperitoneally to the rats |
23 |
|
|
Extracted from the Chaetoceros calcitrans |
Deal with HepG2 cancer cell |
24 |
|
|
Extracted from the brown sea algae L. japonica (kombu) |
Deal with HepG2 cancer cell |
25 |
|
|
Extracted from the brown sea algae Undaria pinnatifida (Wakame), provided by the company of Wuhan Heli |
Deal with Human gastric adenocarcinoma MGC-803 cells |
28 |
|
|
Extracted from the Japanese brown algae Undaria pinna- tifida, supplied by Nippon Seisakusho Co. Ltd. (Beijing, China) |
Deal with Human gastric cancer SGC-7901cells |
29 |
|
|
Purchased from Sigma-Aldrich (St. Louis, MO, USA) |
Deal with human leukemia cell lines, K562 and TK6 |
30 |
|
|
It did not publicly identify the source |
Deal with Human Promyelocytic Leukemia Cells |
31 |
|
Antitumor |
Extracted from the freeze-dried powder of marine alga, Undaria pinnatifida and Codium fragile, respectively |
Deal with Human umbilical vein endothelial cells (HUVECs)
|
32 |
|
|
Extracted from U pinnatifida |
Deal with lymphatic endothelial cells (LEC) |
33 |
|
|
Purchased from Sigma-Aldrich (St. Louis, MO, USA) |
Deal with human MCF-7 breast cancer cells |
34 |
|
|
Purchased from Wako Chemicals (Richmond, VA, USA) |
Deal with human breast cancr line MCF-7 and MDA-MB-231 |
35 |
|
|
Purchased from Sigma-Aldrich (St. Louis, MO, USA) |
Deal with Human leukemia cancer cell line U937 and K562 |
37 |
|
|
Purchased from Sigma Chemical Company (St. Louis, MO) |
Deal with human glioma cancer cell line U87 and U251 |
38 |
|
|
Extract from the edible seaweed Undaria pinnatifida |
Deal with human colon cancer cell lines, Caco-2, HT-29 and DLD-1 |
42 |
|
|
Purchased from the Oryza Oil & Fat Chemical Co. Ltd. (Aichi, Japan) and Dr. Hayato Maeda (Hirosaki University, Japan) |
Deal with azoxymethane-dextrane sodium sulfate (AOM/DSS) carcinogenic model mice |
43 |
|
|
Extract from the chloroplasts of brown sea- weeds |
Deal with human cervical cancer cell lines HeLa, SiHa, and CaSki |
45 |
|
|
Purchased from Sigma- Aldrich (St. Louis, MO) |
Deal with Human NPC cell line C666-1 |
47 |
|
|
Extract from Laminaria japonica |
Deal with human non-small lung cancer cell |
48 |
|
|
Extract from Undaria pinnatifida |
Deal with A549, DLD-1, H1299, MCF7, MDA-MB-231, MRC5, SKOV3, TIG-3, and U2OS cell lines |
49 |
|
|
Extract from dried powder of seaweed (Undaria pinnatifida) after removing carbohydrate and protein was obtained from Riken Vitamin (Tokyo, Japan) |
Fed with Male Wistar rats and female KK-Ay mice |
53 |
|
|
Extract from the P. tricornutum strain UTEX 640 (SAG 1090-1b),which was obtained from the culture collection of Algae (SAG) from the University of Goettingen (Germany) |
Fed with C57BL/6J mice |
54 |
|
Anti-Obesity Effect |
Purchased from Sigma-Aldrich (St. Louis, MO, USA) |
Deal with FL83B cell line |
55 |
|
|
Purchased from Sigma–Aldrich (St. Louis, MO, USA) |
Deal with 3T3-L1 preadipocyte cell |
56 |
|
|
Purchased from Sigma-Aldrich (St. Louis, MO, USA) |
Deal with 3T3-L1 preadipocyte cell |
57 |
|
|
Purchased from Beijing HFK Bioscience Co., 90 Ltd |
Fed with BALB/c mice |
59 |
|
|
Dried Undaria pinnatida was supplied by Jiayi Aquatic Products Co. Ltd. (Shandong, China) |
Fed with C57BL/6J mice |
60 |
|
Effect on the intestinal flora |
Extract from Undaria pinnatifida |
Fed with Escherichia coli and lactobacilli |
61 |
|
|
Extract from D. anceps (wet) at the Punta Plaza location—Antarctic Continent |
Deal with 3T3 mouse fibroblast |
64 |
|
Reduction in oxidative stress |
Purchased from Sigma-Aldrich (St. Louis, MO,USA) |
Deal with raw 264.7 cell |
65 |
|
|
Purchased from Shandong Jiejing Group Corporation (Rizhao, Shandong, China) |
Fed with Male ICR mice |
67 |
|
|
Purchased from Santa Cruz Biotechnology (Santa Cruz, CA, USA) |
Deal with the human keratinocyte cell line HaCaT |
71 |
|
|
purchased from Sigma (St. Louis, MO) |
Deal with LX-2 cells |
73 |
|
Anti-fibroti |
purchased from Biopurify (Chengdu, China) |
Deal with Glomerular mesangial cells (GMCs) |
79 |
|
|
Extract from dried Chaetoceros calcitrans |
Deal with the human liver cancer cells (HepG2) |
85 |
|
|
Extract from Conticribra weissflogii ND-8 |
Deal with C57BL/6 mice and RAW 264.7 cells |
86 |
|
Anti-inflammatory |
Purchased from Sigma-Aldrich (St. Louis, MO,USA) |
Deal with Raw 264.7 cell |
88 |
|
|
Purchase from HiQ Marine Bio- tech Company in Taiwan |
Deal with 70 patients who visited the outpatient department of Gastroenterology and Hepatology, with ages ranging from 20 to 75 years old |
94 |
|
|
Purchased from Sigma-Aldrich (St. Louis, MO, USA) |
Deal with human immortalized keratinocytes HaCaT |
95 |
|
|
Purchased from Sigma-Aldrich (St. Louis, MO, USA) |
Deal with The THP-1 human monocytic leukemia cell line and HaCaT human keratinocytes, and fed with female Swiss CD-1 mice |
96 |
|
Anti-dermatitis |
Purchased from Sigma-Aldrich (St. Louis, MO, USA) |
Fed with male SD rats |
97 |
|
|
Purchased from Phytolox (Okinawa, Japan)
|
Deal with WEHI-3 cells and fed with male Nc/Nga mice |
98 |
|
|
Extract from Undaria pinnatifida
|
Deal with Transfected Chinese hamster ovary (CHO) cells, rat basophil leukemia (RBL) cells, U373 cells, and BA/F3 cells |
100 |
|
|
Purchased from Sigma-Aldrich (St. Louis, MO, USA) |
Deal with Rat pheochromocytoma cells PC-12 |
101 |
|
Other activities |
Extract from Laminaria japonica and purchase from Hi-Q Marine Biotech International Ltd. (New Taipei City, Taiwan) |
Fed with male Sprague Dawley (SD) rats |
102 |
|
|
Purchased from Sigma-Aldrich (St. Louis, MO, USA) |
Fed with Male ICR mice |
103 |
Reviewer 2 Report
Your review summarizes pharmacological activities, especially anti-cancer activities. However, there are few novel points in your review. In addition, in the review of fucoxanthin, its metabolism in the body is very important. Almost of fucoxanthin is metabolized to fucoxanthinol and amarouciaxanthin A. Therefore, author should consider activities of fucoxanthin metabolites as well as fucoxanthin. There are many format mistakes in your MS including references.
Minor comments
L28, “Fucoxanthin is mostly light yellow to brown”? unclear. I think that fucoxanthin is dark orange, but not light yellow.
L31, “below” Page of fucoxanthin structure is changed, but not below. “as shown Figure1” is better.
L32, "ethylenic bonds" is unpopular expression.
L223, HL-60 cell is human leukemia cell line, but not glioma.
Author Response
Dear reviewer,
At first, thank you very much for your advice on this article. Secondly, I would like to express my apologies for the imperfect writing of this article, which has brought you a bad experience in reading. The following is my reply to your suggestions, if there are any incorrect points, I hope you can point them out and I will correct them in time.
Kind regards,
Han Xiao Shan Dong University.
Point 1: L28, “Fucoxanthin is mostly light yellow to brown”? unclear. I think that fucoxanthin is dark orange, but not light yellow.
Response 1: Line 28, this error has been corrected.
Point 2: L31, “below” Page of fucoxanthin structure is changed, but not below. “as shown Figure1” is better.
Response 2: Line 31, this error has been corrected.
Point 3: L32, "ethylenic bonds" is unpopular expression
Response 3: Line 32, the “ethylenic bonds” has been modified to “olefinic bond”.
Point 4: L223, HL-60 cell is human leukemia cell line, but not glioma.
Response 4: Line 223, this error has been corrected. This is a typo, and my intention is to review what has been said in the previous paragraph to introduce what follows.
Round 2
Reviewer 1 Report
The manuscript now entitled “Advances in studies on pharmacological activities of fucoxanthin” has greatly improved on the previous version of the manuscript and I’m happy that the authors made the effort to turn it into a very intersting summary of the current knowledge on fucoxanthin activities. There are minor spelling errors and lack of spaces after sentences’ stop mark. I’ve highlighted some of the words that seem to be misspelled. All species names should be in italic.
Section 3.8 should be revised and shortened. The caption of Figure 2 should be “ Chemical structure of fucoxanthin derivatives fucoxanthinol and amarouciaxanthin A.
The reference list should be revised and formatted according to MDPI guidelines.
Minor comments:
Title – I would insert “the” between “on” and “pharmacological” rendering “Advances in studies on the pharmacological activities of fucoxanthin”
Line 61 - As shown in Table 1 below is a summary of the sources of fucoflavin in the specific experiments I mentioned in this paper.
i - The sentence may be more to the point by inverting the sentence:
A summary of the sources of fucoxanthin in pharmacological studies is shown in Table 1.
ii – I replaced the name of “fucoflavin” by “fucoxanthin”. I don’t know if you have a reason to refer to “fucoflavin” and I think you mean “fucoxanthin”, it is true?
iii – e.g. “Fed with Male ICR mice” should be written as “fed to male ICR mice”
Table 1 it self should be reduced to include the source of fucoxanthin (algal extract or…). As it is referenced, the readers may find the supplier in the referenced publication.

Author Response
Dear reviewer,
At first, thank you very much for your advice on this article. Secondly, I would like to express my apologies for the imperfect writing of this article, which has brought you a bad experience in reading. The following is my reply to your suggestions, if there are any incorrect points, I hope you can point them out and I will correct them in time.
Piont 1 : The manuscript now entitled “Advances in studies on pharmacological activities of fucoxanthin” has greatly improved on the previous version of the manuscript and I’m happy that the authors made the effort to turn it into a very intersting summary of the current knowledge on fucoxanthin activities. There are minor spelling errors and lack of spaces after sentences’ stop mark. I’ve highlighted some of the words that seem to be misspelled. All species names should be in italic.
Response 1 : I'm really sorry for the errors in the format and spelling. I have made a thorough and careful revision, please review it.
Point 2 : Section 3.8 should be revised and shortened. The caption of Figure 2 should be “ Chemical structure of fucoxanthin derivatives fucoxanthinol and amarouciaxanthin A.
Response 2 : I have changed the title of the picture. But I am very sorry, this is the last paragraph I conducted according to the requirements of the other reviewers supplemented revision, he was interested in fucoxanthinol metabolites comparison, it was his hope that I can increase, and he also said that after I submitted the content is more appropriate, so I'm trying not to modify the content of cases to streamline this paragraph, I hope you can satisfied.
Point 3 : The reference list should be revised and formatted according to MDPI guidelines.
Response 3 : The format of the reference has been modified
Point 4 : Title – I would insert “the” between “on” and “pharmacological” rendering “Advances in studies on the pharmacological activities of fucoxanthin”
Response 4 : I have modified the title of the article
Point 5 : Line 61 - As shown in Table 1 below is a summary of the sources of fucoflavin in the specific experiments I mentioned in this paper.
i - The sentence may be more to the point by inverting the sentence:
A summary of the sources of fucoxanthin in pharmacological studies is shown in Table 1.
Response 5 : I have modified this sentence
Point 6 : I replaced the name of “fucoflavin” by “fucoxanthin”. I don’t know if you have a reason to refer to “fucoflavin” and I think you mean “fucoxanthin”, it is true?
Response 6 : I am very sorry, this is a spelling mistake, I have corrected it
Point 7 : “Fed with Male ICR mice” should be written as “fed to male ICR mice” . Table 1 itself should be reduced to include the source of fucoxanthin (algal extract or…). As it is referenced, the readers may find the supplier in the referenced publication.
Response 7 : I have corrected the wrong English expression in the table, but as for the source of fucoxanthin, some of them were purchased from the company, and some were extracted by myself, without purchase channel, so I can only indicate according to the actual source, I am really sorry for that.

Reviewer 2 Report
Dear Author
As I mentioned in the previous review, almost of fucoxanthin is metabolized to fucoxanthinol and amarouciaxanthin A in the body. Therefore, author should consider activities of fucoxanthin metabolites such as fucoxanthinol and amarouciaxanthin A in review paper of fucoxanthin. Only fucoxanthin information against cancer cells in vitro assay is not very useful information for its application. Please consider several papers such as below.
Oncology Letters, 10, 1463-1467, 2015.
Author Response
Dear reviewer,
At first, thank you very much for your advice on this article. Secondly, I would like to express my apologies for the imperfect writing of this article, which has brought you a bad experience in reading. The following is my reply to your suggestions, if there are any incorrect points, I hope you can point them out and I will correct them in time.
Response 1: I'm very sorry that I didn't give you a specific answer before. This time, I have searched the literature with the two derivatives of fucoxanthin metabolism proposed by you, fucoxanthinol and amarouciaxanthin A. By referring to their relevant literature, I have supplemented my paper in this aspect. However, after reading the article you recommended to me, I found that the main reference was fucosterol extracted from algae, which did not seem to belong to the metabolic derivatives of fucoxanthin, so I did not continue to summarize the specific activity of fucosterol, please forgive me. The details are as follows, and you can also see them in the revised article. The image does not show well in the text box, you can view the Word file.
3.8The main metabolites and their activities of fucoxanthin
In addition to the obvious anti-cancer effect of fucoxanthin, its derivatives after metabolism in the body have also become a new research focus in recent years. In recent years, studies have shown that it can be esterified or hydrolyzed to fucoxanthinol in the gastrointestinal tract and further converted into amarouciaxanthin A in the liver [104]. Specific experiments have confirmed that fucoxanthin and its derivatives fucoxanthinol and amarouciaxanthin A can be measured in the plasma, red blood cells, liver, lung, kidney, heart, spleen and adipose tissue of mice after oral administration of fucoxanthin. And through experiments, it is clear that fucoxanthin accumulates in the heart and liver in the form of fucoxanthinol, and accumulates in fat in the form of aurixanthine A in adipose tissue [105]. This result was also verified in another experiment after feeding 0.2% fucoxanthin to diabetic/obese KK-Ay and normal C57BL/6J mice, fucoxanthin accumulates in the form of p-methoquinone in the mice, and fucoxanthinol and amarouciaxanthin A are also detected [106]. In the follow-up experiments of Hashimoto Takash, they found that fucoxanthinol was detected in human plasma, but amarouciaxanthin A, a liver metabolite of fucoxanthin could not be detected. This also indicates that the bioavailability and metabolism of fucoxanthin are different between human subjects and mice [107]. All these indicate that the biological activity of fucoxanthin can be metabolized into fucoxanthinol and amarouciaxanthin A to exert its effects.
In terms of the specific structures of these two metabolites, the structure of fucoxanthinol is similar to that of fucoxanthin. There is a chemically active 5,6-epoxy and non-reactive 5,6-epoxy at both ends of its rigid all-trans long chain. Due to the special epoxy and propylene bond structure, fucoxanthinol have a variety of biological activities, especially in anti-cancer activity, even stronger than fucoxanthin [108]. This activity has also been confirmed by related literature, fucoxanthinol also has significant therapeutic effects on colon cancer cell lines such as Caco-2, WiDr, SW620, HCT116 and DLD-1 [109]. At the same time, amarouciaxanthin A has also been confirmed to be fully synthesized through experiments, through the stereoselective Wittig reaction of C15-allenic and C15-acetylenic tri-n- butylphosphonium salts with the unprecedented C25-3,8-dihydroxy-5,6-epoxyapocarotenal have been completed [110], the specific molecular structure of the fucoxanthinol and amarouciaxanthin A is shown in Figure 2. In addition, the two metabolic derivatives of fucoxanthin also have a variety of activities. According to literature reports, fucoxanthinol and Amarouciaxanthin A in 3T3-L1 adipocytes could show good anti-inflammatory effects by down-regulating the mRNA expression of pro-inflammatory mediators and chemokines induced by RAW264.7 cells [106]. Moreover, it was found in previous reports that fucoxanthin has the effect of anti-obesity and inhibiting the differentiation of 3T3-L1 adipocytes. In the specific study of Mi-Jin Yim et al., it was found that this is mainly due to the fucoxanthin metabolism to fucoxanthinol and Amarouciaxanthin A, thus playing a further role. Fucoxanthinol can down-regulate PPAR (1) mRNA expression, while amarouciaxanthin A can down-regulate PPAR(1) and C/EBPr mRNA expression to inhibit the differentiation of 3T3-L1 adipocytes. In addition, they investigated the regulatory effects of amarouciaxanthin A on the mRNA and protein expression of transcriptional factors, PPARγ and C/EBPR, and on mRNA expression of markers for adipocyte differentiation, it is found that the inhibitory effect of the amarouciaxanthin A is more dominant [111]. In summary, we can understand that not only fucoxanthin, but also its metabolites have various activities that are worthy of our in-depth study and exploration.
Fucoxanthinol
Amarouciaxanthin A
Figure 2. The molecular formula of fucoxanthinol and amarouciaxanthin A

Round 3
Reviewer 2 Report
Dear Author
Thank you for your early revision. I think that it became acceptable MS.
Author Response
Dear reviewer,
Thank you very much for your valuable comments on my article. I feel very happy and grateful for your recognition.
Kind regards,
Han Xiao Shan Dong University.